# Survey mode and nonresponse bias: A meta-analysis based on the data from the international social survey programme waves 1996–2018 and the European social survey rounds 1 to 9

**Adam Rybak** *

Faculty of Sociology, Adam Mickiewicz University, Poznań, Poland

* adam.rybak@amu.edu.pl

## Abstract

The constant increase in survey nonresponse and fieldwork costs are the reality of survey research. Together with other unpredictable events occurring in the world today, this increase poses a challenge: the necessity to accelerate a switch from face-to-face data collection to different modes, that have usually been considered to result in lower response rates. However, recent research has established that the simple response rate is a feeble measure of study quality. Therefore, this article aims to analyze the effect of survey characteristics, especially the survey mode, on the nonresponse bias. The bias measure used is the internal criteria first proposed by Sodeur and first applied by Kohler. The analysis is based on the survey documentation and results from the International Social Survey Programme waves 1996–2018 and the European Social Survey rounds 1 to 9. Random-effects three-level meta-regression models, based on data from countries from each inhabited continent, were created in order to estimate the impact of the survey mode or modes, sampling design, fieldwork experience, year of data collection, and response rate on the nonresponse bias indicator. Several ways of nesting observations within clusters were also proposed. The results suggest that using mail and some types of mixed-mode surveys were connected to lower nonresponse bias than using face-to-face mode surveys.

## Introduction

The constant increase [1, 2] in survey nonresponses and, thus, possible nonresponse errors create difficulties with the implementation of the survey method. While in some countries this issue is addressed by increasing so-called "fieldwork effort" [3 p101–113], resulting in differing outcomes, in many others the skyrocketing costs and problems associated with gaining access to specific populations result in switching from "traditional" face-to-face surveys to other modes. The consequences of the COVID-19 pandemic amplified this changing approach. In many countries that still deemed standard face-to-face collection to be optimal, the law

**Data Availability Statement:** All data are available at OSF repository: Dataset 10.17605/OSF.IO/VTWN7 R script 10.17605/OSF.IO/QM5RU Data

citations 10.17605/OSF.IO/D6PZG SPSS syntaxes
for NBI calculation 10.17605/OSF.IO/JKE5H
Software citations 10.17605/OSF.IO/RX7Q5.

**Funding:** The author gratefully acknowledges
funding from the National Science Centre, Poland
(AR: PRELUDIUM-17 grant no. 2019/33/N/HS6/
00322 https://ncn.gov.pl/en). The funder had no
role in study design, data collection, analysis,
decision to publish, or manuscript preparation.

**Competing interests:** The author has declared that
no competing interests exist.

necessitated a change, which raises the question: Should we treat this situation as a risk or an opportunity? To reduce uncertainty in this subject, the following study aims at estimating the impact of using different modes of data collection with control of the variables, such as the sampling design, survey program, institutional experience, and response rate (RR), on survey representativeness. I treat these variables as controls because, for example, sampling design is usually predetermined by accessible sampling frames, while the survey mode used is (with limitations) a matter of diligent choice. As the data are hierarchical, differences between countries and program waves are also accounted for.

There is already quite a substantial body of survey methodology literature concerning relations between the survey mode and sampling on the one hand and nonresponse on the other. Unfortunately, there are far fewer studies that consider nonresponse bias. Additionally, most methodological analyses are based on more fragmented sources, such as single surveys or a few journal article meta-analyses. Exceptions [e.g., 2] usually lack the implementation of net sample quality indicators other than nonresponse rates. In most cases, they also stick only to European or US data, while there is a substantial tradition of doing non-face-to-face surveys in countries like Canada, Australia, and New Zealand. For this reason, I have based this study on the reports and data from both the European Social Survey (ESS) rounds 1 to 9 and the International Social Survey Programme (ISSP) waves 1996–2018. The reason is that ISSP surveys present a very high variability of survey modes compared to other international programs. Yet the limitation of this study is the usage of observational data; modes are not assigned randomly to surveys. To mitigate this issue a little, fully face-to-face ESS surveys have been added to the database to provide parallel surveys conducted in the same countries in the same timeframe, at least for a portion of cases. While some studies [4, 5] suggest that the quality of ISSP documentation is inferior to that of the ESS, it is still one of the better documented international survey programs today.

## Nonresponse bias

Unit nonresponse is a major component of total survey error, which itself is currently the default framework for conducting survey methodology research [6–8]. The other types of error are not a subject of this study, nor will I delve into other issues connected to cross-national comparisons. Measurement error or problems with survey research costs will only be mentioned. As for nonresponse itself, if not connected to systematic error, it will pose a threat only to the precision of the estimates. On the other hand, "nonresponse bias results from the contrast between those responding and not responding to the surveys" [9]. Nonresponse bias threatens the representativeness of a survey. For a formal presentation: [e.g., 3 p29–32].

There are many possible ways of estimating the size of nonresponse bias. The most common in survey practice—but highly criticized [10–13] for establishing only a higher bound of the possible error and having a limited relation with other measures—is the RR. As for the other most feasible, possible measures of survey representativeness, they can be divided into two categories. The first category requires individual-level auxiliary data about a whole drawn sample. The most popular members of this category are so-called R-indicators [14–16]. Unfortunately, such required auxiliary data are not available for this study. The second category, which does not require such data, can be divided into the two most popular approaches. The first one, sometimes called "external criteria of representativeness," requires access to reliable population statistics [9, 17–19]; they can be obtained both from official statistics and other survey data. This approach would not be optimal for an assessment concerning survey data covering such a wide geographical area as the ISSP. Official data or other survey data can and do strongly differ in quality between countries (not to speak of continents). Therefore,

differentiation between the effects of nonresponse bias and bias coming from faulty population statistics would be problematic.

As for the second approach, Sodeur [20] proposed a way of estimating the quality of a net sample based on the so-called "internal criteria of representativeness." It was then developed by Kohler [21] and used in several subsequent studies [e.g., 17, 22–24]. This method is based on finding the proportion in the sample, which is known *ex definitione*. Souder's example is heterosexual couples living together in two-person households; the gender balance in this situation should be 50–50. "Internal criteria assessments probe the integrity of the sample by exploiting this a priori knowledge and point to potential irregularities in the survey process" [25]. Kohler [21] argues that this measure would not be affected by coverage and observational errors nor by item-nonresponse (unfortunately only for face-to-face surveys, where gender is in most cases assessed by the interviewer), making it a good candidate for the role of assessing survey representativeness. Ideally, it would be possible to control for same-sex couples, yet this is rarely the case (surely not for the ISSP data, crucial for this study). However, the working assumption could be made that there are no more or no fewer gay couples than lesbian couples in the population. For ESS rounds 1 to 8 and European Quality of Life Survey rounds 1 to 4, where controlling for same-sex couples is possible, the correlation between indicators using strict (with control) and lenient (as in this paper) approaches to assess nonresponse bias was 0.96 and 0.93, respectively [25]. The impossibility of using a strict approach is an important limitation of this study.

Additionally, counting only official marriages would make a big difference between countries with different legal landscapes regarding this matter, so both formal and informal partnerships should be included. In the following parts of this paper, I will refer to Sodeur's internal criteria of nonresponse bias assessment as a nonresponse bias indicator (NBI).

For an extensive discussion on the usability of the different nonresponse bias measures and indicators in similar research, see the work of Jabkowski, Cichocki, and Kołczyńska [25]. Additionally, it must be stressed that, in reality, nonresponse bias is a property of a specific variable. It can have different sizes for different survey questions. For this reason, the measure used here must be treated only as a proxy. There are some additional possibilities for assessing the quality of the survey in terms of the nonresponse bias. One could use the gross sample information if it consists of individuals with known characteristics, but this is possible only in the case of individual samples [26]. Additionally, nonrespondents could be targeted by very short follow-up questionnaires, as in the ESS, but such follow-up may also be vulnerable to nonresponse. A similar vulnerability is shared by any method that compare early and late respondents, or individuals who needed the conversion procedures with the rest of the respondents [19]. Such assessment methods are only possible for the secondary data analysis, when required information is easily accessible to external researchers, which is rarely the case.

## Survey modes

The face-to-face mode has usually, especially in the past, been considered the best mode for achieving high RRs [27, 28] or considered the "gold standard," which is reflected in the mode most commonly used in the most important comparative survey programs. For example, the Eurobarometer, ESS, and European Quality of Life Survey waves conducted in pre-COVID-19 times (i.e., before 2020) were based solely on the face-to-face mode. Moreover, in the case of the World Values Survey and the European Values Study, this mode was clearly dominant. The rationale for considering this mode the best may be the possibility it provides for the interviewer to use many motivational and conversion procedures and to gather paradata. It can also be driven by the mode preference of potential respondents. Due to very limited research,

preference for face-to-face surveys among respondents may be the most widespread, at least in some countries [29, 30]; but apparently, declared preference also depends on the mode of the question administration if the preference is studied through declarative, not behavioral, data [first observed by 31]. I will not discuss the possible causes of these preferences, whether are they psychological [32], cognitive [33], or of another type. The widespread usage of this mode in scientific surveys may also come from the lack of the sampling frame necessary to use any other mode, but this is mostly the case in low- and middle-income countries. On the other hand, the main downside of this survey mode is its cost—usually the highest of all other single-mode surveys [34]. Moreover, the high RRs it generates may stem, if proper supervision and backchecking is not implemented, from fraudulent activity on the part of the interviewer.

Mail surveys have commonly been considered as generating lower RR than face-to-face surveys [28, 35–37]. On the other hand, they are treated as a way to overcome the social desirability bias present mainly in the face-to-face context [38, 39]; however, this type of bias is not a focus of this study. The downside of the mail mode is that it becomes more problematic if no individual frame is available. Random within-household selection made by household members may be of lower quality than one made by a skilled interviewer. However, the RR of mail surveys may also be higher than those of current web surveys [40, 41]. Nevertheless, there are multiple examples of well-designed mail surveys that achieved similar or even higher RRs than one would expect from face-to-face surveys in a similar context [42 p351–352].

While mixed-mode surveys in a very broad sense were part of survey practice from the early years (when self-administered questionnaires started to be used for asking sensitive questions in face-to-face surveys) and started to be more widely discussed in the 1960s, they became very popular only in the 21st century [43]. The idea of mixing different modes of data collection is motivated mainly by the desire to reduce cost, social desirability bias, or nonresponse (but not all at once). It may also involve modes of contact and a follow-up phase. Mix-mode surveys could possibly be cheaper than face-to-face surveys while achieving higher RR if carefully designed [44]. I will not discuss the complex differences between various types of mixed-mode [42 p398–449, 45–47] implementation: sequential and parallel design, mode choice [48], mode preference [29, 49–51], adaptive or responsive design [52]. It is because the mixed-mode surveys I will analyze in the forthcoming parts of this paper are heterogeneous in implementation specifics, as are single-mode surveys. In a study consisting of 795 cases, it would be impossible to address these differences substantially.

As stated previously, a body of literature concerning relations between survey mode and nonresponse bias exists but is not very large; nevertheless, any endeavor aimed at finding clear conclusions in it may be fruitless. This paper is a systematic review, not a narrative one. Therefore, the list of findings is more of an illustration. I will start with the single-mode comparisons. De Leeuw [28] in her experiment found that only face-to-face surveys (in comparison to mail and computer-assisted telephone interviewing, CATI) generated significant differences between gender proportion in the sample and the register data. Laaksonen and Heiskanen [53] found in face-to-face, phone, and web parallel surveys in Finland that the face-to face mode was the least representative as measured by R-indicators. Klausch, Hox, and Schouten [54] found in their mode experiment comparing face-to-face, CATI, web, and mail modes that face-to-face and web modes achieved the best representativeness measured by R-indicators. There are also some comparisons between single-mode and mixed-mode surveys. Fowler et al. [55] found results from a mail and CATI mixed-mode survey more representative than from a single-mode mail survey. Dillman et al. [56] showed that using a mixed-mode survey design can potentially raise RR compared to a single-mode survey, but only non-face-to-face modes were included in this study; the authors did not find any significant differences in representativeness between single-mode and mixed-mode approaches. Luiten and Schouten [57]

compared a single-mode CATI survey to a mixed-mode web, mail, and CATI survey; while both had similar RR and cost, the mixed-mode approach was more representative as measured by R-indicators. Klausch, Hox, and Schouten [54] compared single-mode CATI, mail, and web surveys with their respective mixed-mode approaches with added face-to-face follow-up; in most cases, the bias measured by R-indicators for mixed-mode surveys was lower. Bianchi et al. [58] found that switching the mode in the panel survey from a face-to-face to a mixed-mode approach resulted in similar RR and sample composition while lowering the cost. Villard and Fitzgerald [34], after analyzing a series of mode experiments conducted under the auspices of the ESS, concluded that "mixed-mode data collection designs are unlikely to increase RR or reduce nonresponse bias in survey estimates compared to data collected face to face. In fact, overall, they suggest that RR would decrease and nonresponse bias would probably increase." Cornesse and Bosnjak [59], after conducting a meta-analysis of 96 surveys, found that using mixed-mode surveys resulted in nonresponse bias reduction measured by R-indicators compared to single-mode surveys. Sztabiński [60], in an ESS mode experiment in Poland, found that using a mail, web, and face-to-face mixed-mode survey resulted in lower RR, higher costs, and a similar sample composition as a parallel single-mode survey. Wolf et al. [44], after an analysis of the 2017 round of the European Values Study, found that face-to-face surveys were more representative than some mixed-mode surveys but that the latter were much cheaper and had higher RR.

The preceding section offered an extensive overview of the current state of knowledge about the relationship between survey modes and various propositions of measures of survey representativeness. While an important body of research exists in this field, the results can be difficult to reconcile due to the limitations of traditional forms of narrative review or even such approaches as simple vote counting of significant results. These limitations can lead readers to gain a false impression of conflicting results in many fields [61 p20]. To address these issues, this study endeavors to shed more light on this subject by harnessing the potential of meta-analysis, which is a "statistically defensible approach to synthesizing empirical findings" [62 p8]. Therefore, this study aims to answer the following research questions: What is the relation between survey mode and survey representativeness, and how is this relation affected by other survey characteristics and broader context?

## Data and methods

The data for this study come from methodological reports and data from ISSP waves 1996–2018 and ESS rounds 1 to 9. All ISSP documentation and datasets are stored and publicly available in the GESIS–Leibniz Institute for the Social Sciences repository [63]. The ISSP is a cross-national survey research collaboration between institutions from 57 countries from all inhabited continents, beginning in 1984 [64]. While being decentralized and differing highly in the survey modes used and fieldwork procedures, it maintains high standards of unified reporting and quality control [65]. ESS documentation and datasets are available on the official ESS site [66], while the Norwegian Centre for Research Data provides hosting. The ESS is one of the most sophisticated methodologically among big cross-national comparative social survey programs [67]. It started in 2001, inspired by the growing discontent of the academic community for previous comparative surveys [68]. All nine pre-COVID-19 rounds of the ESS were guided by a set of ambitious principles: "non-duplication, longitudinality, cross-national equivalence, potential for multilevel analysis, ongoing refinement of methodology, simultaneous flexibility and stability, easy and inexpensive data access and clear definitions of both population and sampling" [69]. Moreover, as stated previously, all the pre-COVID-19 ESS surveys were done in a face-to-face mode, either as PAPI or CAPI. Data references can be found in the separate file [70].

Reports and datasets were used to code each survey's characteristics as part of a funded research project. For this paper, the most important information was about the program, year and country, sampling design (including within-household selection), use of quotas or substitution of nonresponders, mode (or modes) of the survey, and outcome numbers used to calculate RR. RRs were calculated from the outcome numbers even if the separate value was present in (some of) the reports. Of the six ways of RR calculation proposed by the American Association for Public Opinion Research, I selected RR6 [71], because of some inconsistencies in the reporting of the "unknown eligibility" category in the ISSP.

$$RR6 = \frac{(I + P)}{(I + P) + (R + NC + O)}$$

where I and P represent complete and partial interviews; R represents refusals and break-offs; NC represents noncontacts; and O represents other (eligible) nonresponses. Because ESS outcome reporting is constructed in a different way, personal communication was needed to establish how to calculate RR6. Result datasets were used to code the NBI. The idea of this measure was described previously. NBI, being in this context a single study effect size, was calculated as:

$$ES_i = |prop_{fem} - 0.5|$$

where $prop_{fem}$ is the proportion of females in a subsample of couples living together in two-person households and n is the size of a subsample of couples living together in two-person households [21]. As the proportion on which the $ES_i$ is based is only an estimate and the true proportion of females in the defined subpopulation is already known to equal 0.5, the within-study variance may be calculated as:

$$V_i = 0.25/n_i$$

where $n_i$ is the subsample size of the single study [21, 23, 24]. Information about the "survey experience" (how many times the survey was done before in one country in one program) was calculated afterwards by the author. While obviously not capturing all information about possible fieldwork institution change between waves and the degree of shared experience or field networks between such institutions, it is a proxy for how much information about the local surveying climate and context was available to people conducting such a survey.

Initially, there were 233 coded ESS surveys from rounds 1 to 9 and 760 coded ISSP surveys from waves 1996–2018. Exclusion criteria, meant to compare only surveys using entirely random sampling, were: using nonrespondents substitution, quotas, or nonrandom within-household selection, not providing complete information about sampling, survey mode, outcome numbers, and information needed to compute NBI. After the exclusion, the database consisted of 795 surveys (223 ESS and 572 ISSP). The created dataset is available at [72], and the SPSS syntaxes used for NBI calculations are available at [73]. The exclusion process is described in **Fig 1**.

Because the analysis was conducted on a database created by the author [based on 75], who used all the methodological reports and datasets provided by the repositories formerly listed to create it, and because of the traditions and specific nature of meta-analysis in the survey methodology context, it is not possible nor necessary to use the entire PRISMA standards [76] for reporting, as is done in clinical trial meta-analyses.

As stated previously, all ESS surveys are done in face-to-face mode and in European countries. ISSP surveys are done in multiple modes in countries from all inhabited continents. **Table 1** shows the distribution of modes by the United Nations world regions.

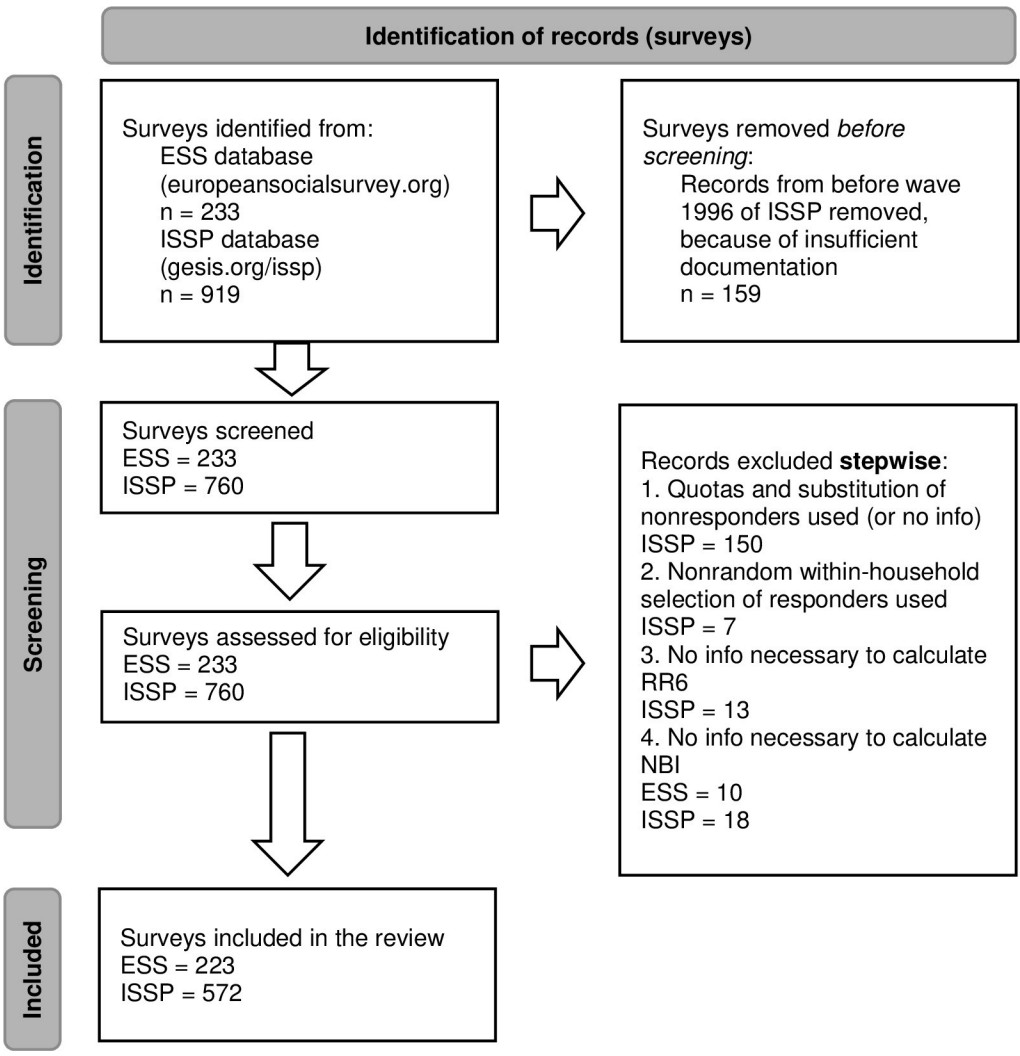

**Identification of records (surveys)**

Surveys identified from:
ESS database
(europeansocialsurvey.org)
n = 233
ISSP database
(gesis.org/issp)
n = 919

Surveys removed *before screening*:
Records from before wave 1996 of ISSP removed, because of insufficient documentation
n = 159

Surveys screened
ESS = 233
ISSP = 760

Surveys assessed for eligibility
ESS = 233
ISSP = 760

Records excluded **stepwise**:
1. Quotas and substitution of nonresponders used (or no info)
ISSP = 150
2. Nonrandom within-household selection of responders used
ISSP = 7
3. No info necessary to calculate RR6
ISSP = 13
4. No info necessary to calculate NBI
ESS = 10
ISSP = 18

Surveys included in the review
ESS = 223
ISSP = 572

**Fig 1. PRISMA flow diagram.** From [74].

As for the categorization of the modes, face-to-face and mail-mode categories included single-mode surveys conducted in the stated mode. ISSP questionnaires are quite often fielded as a self-administered supplement to the face-to-face main survey. Apart from that, sometimes ISSP surveys were reported as being fielded in single-mode self-completion (especially popular in ISSP Japan, 15 surveys); it indicates a self-administered questionnaire delivered and collected by an interviewer. Both of these situations fulfil the criteria for being called a mixed-mode survey [45, 46]. From all 96 ISSP mixed-mode surveys present in the final dataset, where face-to-face was one of the modes, only three contained any survey mode more than face-to-face and self-completion. It was in the year 2000 in Switzerland, where some additional contacts after these two modes were made by telephone and mail, and in the 2012 and 2013 ISSP in Iceland, where face-to-face nonresponders could fill out an online questionnaire. For this reason, all these surveys were included in one category: mixed-mode (with face-to-face). As for the last category—mixed-mode surveys (without face-to-face)—it included (ISSP) surveys fielded in different combinations of mail, web, and CATI survey modes. This type of survey was especially popular in Nordic countries: 15 surveys in Denmark, 10 in Norway, 8 in

**Table 1. Distribution of survey modes in analyzed dataset by the United Nations world regions.**

|  | Face-to-face (of which ESS) | Mixed-mode (with f2f) | Mail mode | Mixed-mode (no f2f) | Total |
|---|---|---|---|---|---|
| Australia and New Zealand | 0 | 1 | 27 | 3 | 31 |
| Caribbean | 4 | 0 | 0 | 0 | 4 |
| Central America | 10 | 0 | 0 | 0 | 10 |
| Eastern Asia | 32 | 20 | 0 | 0 | 52 |
| Eastern Europe | 115 (47) | 9 | 0 | 0 | 124 |
| Northern America | 15 | 8 | 9 | 0 | 32 |
| Northern Europe | 100 (72) | 26 | 39 | 42 | 207 |
| South America | 20 | 2 | 0 | 0 | 22 |
| Southeastern Asia | 16 | 0 | 0 | 0 | 16 |
| Southern Africa | 14 | 0 | 0 | 0 | 14 |
| Southern Asia | 6 | 0 | 0 | 0 | 6 |
| Southern Europe | 93 (38) | 0 | 0 | 0 | 93 |
| Western Asia | 40 (13) | 0 | 0 | 0 | 40 |
| Western Europe | 81 (53) | 30 | 26 | 7 | 144 |
| Total | 546 | 96 | 101 | 52 | 795 |

Finland, and 5 in Iceland. As stated previously and subsequently, ESS face-to-face surveys were included in the model to alleviate issues originating from this study's observational, non-experimental character.

## Meta-analysis

Most commonly, meta-analysis is used to create proper statistical synthesis of results taken from many articles or clinical trials. In that situation, some typically used dependent variables, called effect sizes or treatment effects [77 p17], are usually computed to either statistically standardize the different measures used by different authors or comply with standard settings of experimental studies in medicine. Therefore, readers probably encountered two types of effect sizes: "those based on differences of two means and those based on differences of two proportions" [78 p13]. However, these examples do not exhaust the usefulness of meta-analysis. On the contrary, "effect size is simply the size of anything that may be of interest" [79 p34], whether it is measured in exactly the same way for every study or recalculated to make it so. For example, in epidemiological studies, the true population mean, or proportion, may be of interest [80]. Card proposed that the working definition of effect size could be "an index of the direction and magnitude of association between two variables" [62 p87], a definition that would include the measure used in this paper, if interpreted *sensu largo*.

An important question while doing a meta-analysis is whether to use a fixed- or random-effects model [77 p61–86]. Fixed-effects models are justified only when the studies included are nearly identical and one is not interested in treating them as part of a bigger population. Therefore, random-effects models, being more robust and allowing for some generalizations, should be used in most cases. The simplest meta-analytical random-effects model looks like the equation below:

$$\widehat{OES^*} = \frac{\sum_{i=1}^{k} w_i^* \times ES_i}{\sum_{i=1}^{k} w_i^*}$$

"Parameter k is the number of included studies and estimator $ES_i$ stands for single-study effect size, i.e., the difference between the observed and the true proportion of females in two-

person heterosexual households. For each study, weights $w_i^*$ are computed as reciprocals of the variance of $\widehat{OES^*}$, i.e., $V_i^* = V_i + \tau^2$, where $V_i$ is within-study variance and $\tau^2$ denotes between-study variance" [23, 77].

When enough heterogeneity between studies is found, meta-regression is the usual method of choice to analyze it [81]. To avoid spurious findings while conducting meta-regression, methods such as using the so-called Knapp-Hartung test [82] or permutation test were proposed [83–85]. Nevertheless, these typical two-level models, where the "hidden" first levels are usually individuals and the second levels are study summaries are sometimes not enough [86–89]. When one could expect that studies are nested in some higher-level cluster, one should use the three-level meta-regression model to deal with such dependencies. For discussion on why using more common hierarchical models is not enough, please see [90]. Studies can and often are nested in more than one set of clusters at a time—that is, random factors may be crossed; to address that situation, cross-classified models can be used [91]. Many types of meta-analytical models have already been used to conduct research in the area of survey methodology, all of which either synthesize results from different papers or are based on self-calculated measures for selected survey programs [e.g., 10, 37, 40, 41, 48, 59, 92–98].

The effect size I used for the meta-analysis was, as stated before, NBI. The choice of moderators for the models was based on both state-of-the-art and statistical evaluation. Survey mode and frame constitute probably the most basic set of survey characteristics. The subject of mode was extensively described in the introductory parts of this paper. As for the sampling frame, the frame of individuals is usually considered better in terms of representativeness than other types of frames, primarily because of the burden connected to the process of within-household selection of the respondent at the last sampling stage [22, 99, 100]. Therefore, I used the binary variable individual/nonindividual (household/address, area probability sampling). To not overcomplicate the model, I ignored the possible impact of the within-household selection method on nonresponse bias [96]. Exploratory subgroup analysis and previous knowledge about possible issues connected with the aforementioned respondent selection not being supervised by the interviewer (as in, for example, mail mode) led to the conclusion that the interaction between those two moderators may also be an important factor in explaining differences in NBI values (Readers can conduct all analyses mentioned in the article, and others, using the R script available at [101].) Therefore, mode, frame, and interaction between them were selected for the simplest model specification. The analyzed dataset consists of studies from two survey programs that differ significantly in terms of funding and organization [102, 103], hence the inclusion of the binary variable ESS/ISSP in the extended model. Institutional experience can influence outcome rates of a survey [3 p102–113] and thus at least the upper boundary of possible nonresponse bias. I chose to include a moderator that for each observation simply counts how many surveys were done previously in this country and program. This study, as stated previously, is based on observational data—like many meta-analyses in survey methodology [98]. If one, however, would like to apply the categories of variables usually included in the causal inference models, then it should be noted that this study is focused chiefly on the effect of survey mode on nonresponse bias and that every candidate for the meta-regression moderator listed above should be treated as a confounder. The sampling frame, institutional experience, and survey program can usually have an impact on the choice of mode(s) and separately on the nonresponse bias. As for the last candidate for a moderator—the RR (RR6)—its inclusion can be used to estimate the partial effects of survey modes holding RR constant. In the causal inference model, it would be a mediator. RR6 was included only in the last of the models presented in the results.

After testing if the inclusion of stated moderators is not only theoretically but also statistically sound (addressed in the results section), identification and exclusion of outliers were

performed based on the influence diagnostics of outliers [104] and on quantile-quantile plots [105, 106]. The four identified highly problematic observations included ESS surveys in Slovakia conducted in rounds 4, 5, and 6 and an ESS survey in Lithuania from round 9. All of them were face-to-face surveys with unusually high value of NBI. Lithuanian cases were already identified by Jabkowski, Cichocki, and Kołczyńska [25]; the sampling design switched from one based on the individual register frame in round 3 to an area sample with house enumeration in round 4. "This change dramatically increased interviewer discretion in regard to respondent selection," which resulted in oversampling women. As for the case of Lithuania in round 9, it is a bit more problematic. Between rounds 8 and 9, the data collector changed, the mode switched from PAPI to CAPI, and the Kish grid replaced the last birthday method. Moreover, the DEFFp rose from 1.34 to 1.64 and the number of completed interviews dropped from 2,122 to 1,835 (References to documentation are available at [70].) Nevertheless, none of these, unfortunately, explains such a rise of NBI. The final models after the exclusion of outliers are based on 791 observations.

To answer the research questions, I created random-effects meta-regression (alternatively called mixed-effects) models using the restricted maximum likelihood method to estimate $\tau^2$ [77 p114–117]. Every model was created with the R package metafor [107]. Software citations are available at [108] and the R script for replication at [101]. To address the issue of surveys probably being nested in some third-level clusters, which may result in observation dependency, I introduced cross-classified models [91], where surveys were nested simultaneously in program waves and countries. Surveys conducted on one country's population at different points in time can be treated as repeated observations by some, therefore the need for nesting. Countries can also differ in survey climate, but it is hard to measure, and such measure was not included. Instead, the random intercepts for countries may, to some degree, serve the role of estimating it. The author considered including a variable measuring cultural trust at this level, but the most important, comprehensive source of such were the World Value Survey results, and the idea of basing survey evaluation on data from a methodologically less "diligent" survey was rejected. Surveys being part of specific comparative program waves (counted here separately for both programs) can share many characteristics, such as timeframe, methodological standards, and especially questionnaire topics. Topic saliency can have significant impact on unit nonresponse [109–111], but it is not uniform for the whole population. Therefore, it was accounted for only in the random intercept of waves. The moderator that explained part of wave variance only is the survey program because it does not differ inside the cluster. It is usually considered that there should be at least 20 third-level clusters [112 p46–48]. However, it is a matter of discussion [113] and is not always followed by researchers [86, 114]. In this case, surveys were nested in 32 waves and 53 countries, which is a clearly sufficient number. As the ICC calculation and the checking of $I^2$ distribution between model levels [115] are extremely problematic in cross-classified models, such values were gathered for surveys nested in countries and in waves separately. Cross-nested models with three sets of moderators are presented in the results part. The final two-level model and the three-level model with RR6 as an effect size can be consulted in the S1 Table.

The raw, nontransformed proportions [for transformation proposal, see, e.g., 80] were used to calculate NBI. It is because the standard error of the NBI is based on the known "true" proportion of 0.5; therefore, there was no identified risk for the variance calculation. Moreover, subsamples and proportions were calculated without using weights. As weights are not standardized between countries and in the ISSP most of them are poststratification weights, it is better for comparisons to use raw data [For extensive discussion on this subject, please see 25.] If design weights were available for all analyzed cases, it would be the best choice to use them in calculations.

## Results

Standard, two-level random-effects meta-analysis of surveys using NBI as an effect size identified the mean value of effect (thus, NBI) as 0.0366 (95% CI 0.0341–0.0391). The test of heterogeneity is significant (Q(df = 794) = 1532.43; p<0.0001). $I^2$ ("the percentage of variability among effect sizes that exists between studies relative to the total variability among effect sizes" [62 p188–189]) has a value of 49.15%. After the exclusion of four outliers, the only major change concerned $I^2$, which dropped to 31.71%, a value that could be interpreted as a low-to-moderate level [116], a still sufficient heterogeneity between studies to conduct a meta-regression.

As we can see in **Fig 2**, simple random-effects meta-analysis in subgroups shows that NBI values for mixed-mode surveys with face-to-face (mostly face-to-face with additional self-administered questionnaires) and for the mail mode clearly overlap with the estimate for single-mode face-to-face surveys. At this level, it is not possible to tell whether values of bias estimate for other mixed-mode surveys are smaller than for face-to-face. For context, we can see in **Fig 3** that the values of RR6 for modes using face-to-face surveys in any configuration clearly outperform other ones, which is in accordance with common expectations concerning this subject. Based on the presented literature, it was expected that the connection between realization and bias would be weak at most.

### Meta-regression

Meta-regression based on the full set of observations, with the basic set of moderators (mode*frame), was compared with models including additional variables: program, repetitions, and a set of both. The likelihood ratio test suggests that, statistically, the full set is the best: $\chi^2(2) =$

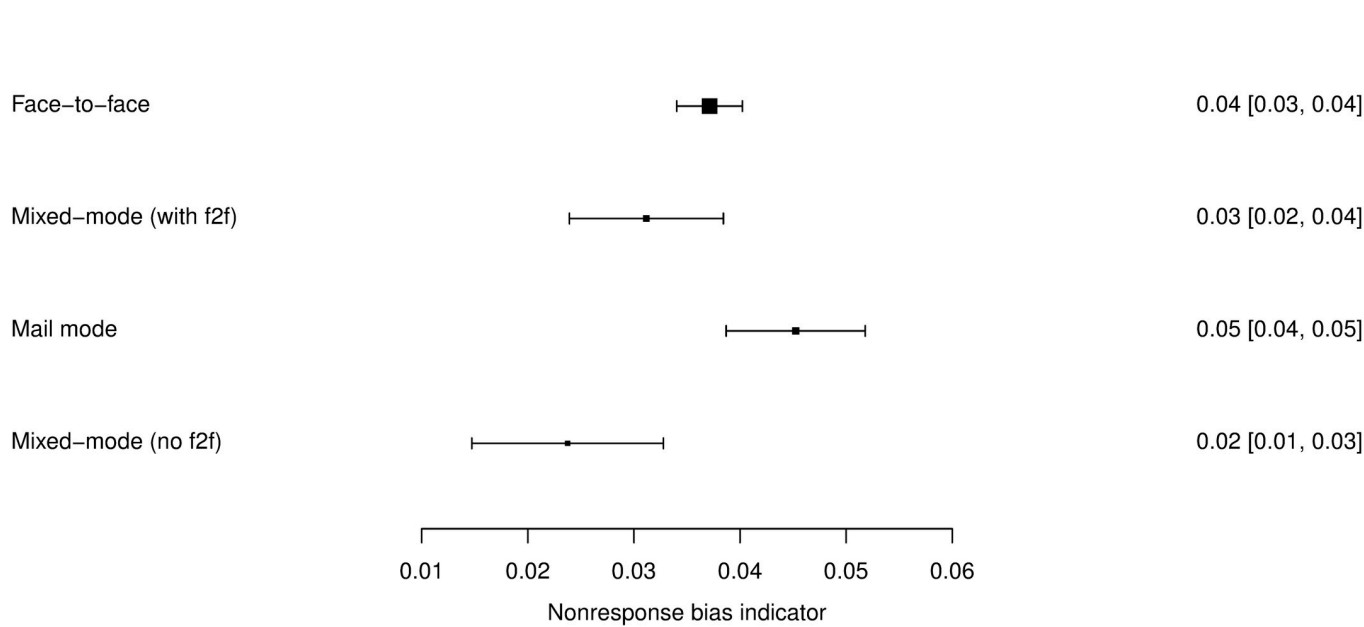

**Fig 2. Distribution of NBI values of surveys in the dataset by survey modes—forest plot.** N = 795.

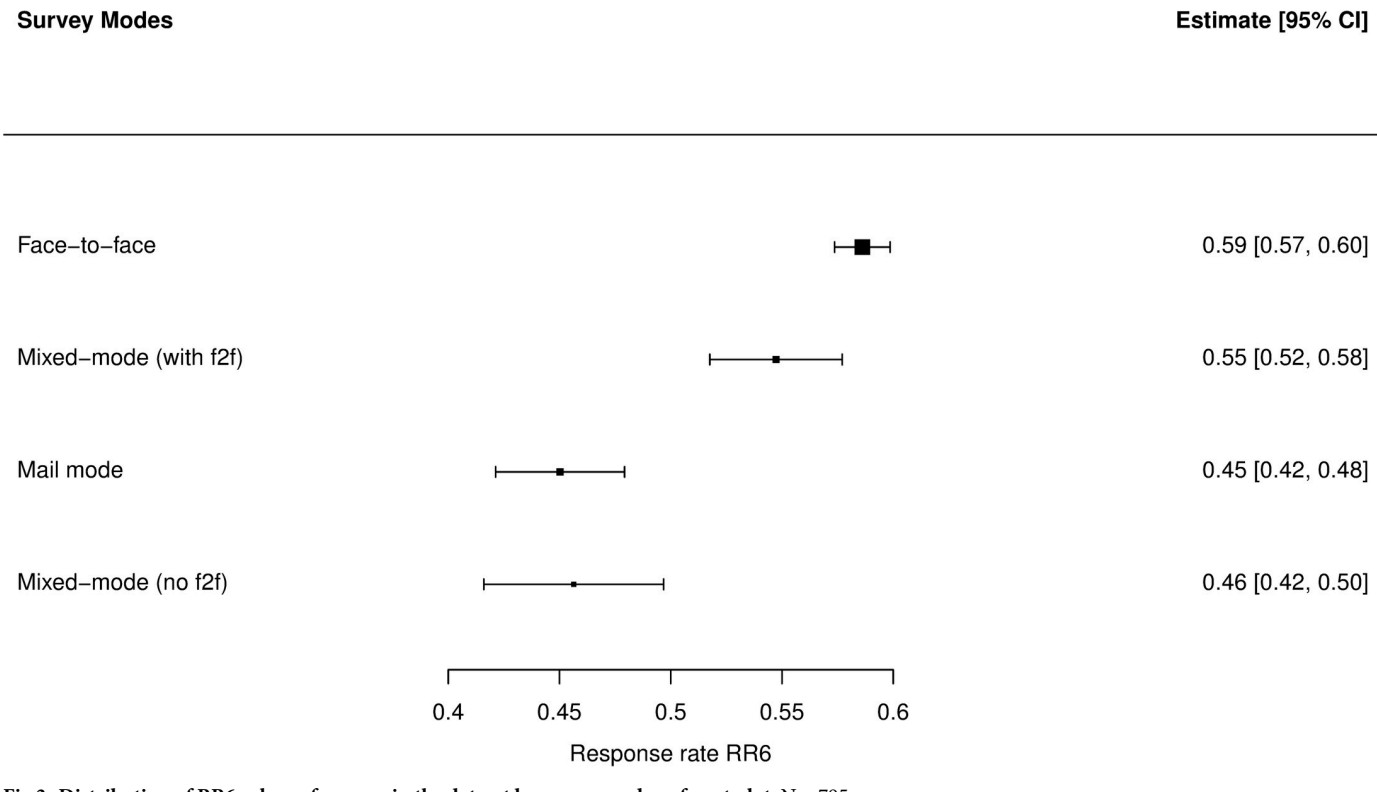

**Fig 3. Distribution of RR6 values of surveys in the dataset by survey modes—forest plot.** N = 795.

13.09;p = 0.001. The model using this set of moderators, restricted maximum likelihood, and the Knapp-Hartung method, with p-values adjusted according to the permutation method, found mail mode, mixed-mode (no face-to-face), frame, program, repetitions, and mail-non-individual frame interaction moderators to be significant. This model was used for quantile-quantile plot and influence diagnostics, which both have resulted in flagging four observations as problematic outliers. These outliers were already described in the data and methods section and were excluded from any further analysis, which lowered the number of observations to 791. The two-level model designed the same as in the previous step, but with outliers excluded, differs in the frame moderator not being statistically significant and the mixed-mode (with face-to-face) starting to be statistically significant, having a pseudo-$R^2$ of 77.52%, and a significant test of moderators: F(df1 = 9, df2 = 781) = 29.0204;p = 0.001. Multimodel inference [117] confirmed that the final set of moderators is optimal. This model is available in S1 Table.

As calculating the ICC and $I^2$ distribution is problematic for cross-nested models, I calculated them for models with country and wave as the third level separately. Every three-level model in this paper uses t- and F-distributions for making inference, and uses "an improved method for approximating the degrees of freedom" [118 p128] for these distributions, as the full Knapp-Hartung method is unavailable, and the models also exclude the four outliers. According to a likelihood ratio test (intercept only), the model with surveys nested in waves (and no moderators) is statistically better than the two-level model: $\chi^2(1) = 5.2$;p = 0.02 and has an ICC value of only 0.1, 28.6% of the total variance at level 2, and only 3.28% at level 3. The model with surveys nested in countries (and no moderators) is statistically better than the two-level model: $\chi^2(1) = 97.64$;p<0.0001 and has an ICC value of 0.65, 13.1% of the total variance at level 2, and 24.1% at level 3. The cross-classified model (surveys nested in countries

and waves) has all three variance components identifiable according to likelihood profile plots [118 p171]. **Table 2** presents a set of three cross-classified models with different sets of moderators. The first model uses the most basic set—mode, frame, and their interaction—and has a pseudo-$R^2$ of 58%. The second model uses the optimal set of available moderators with added program and repetition ("institutional experience") variables and has a pseudo-$R^2$ of 67%. The third model, aimed at estimating the partial effect of the mode holding outcome rates constant, includes an additional moderator: RR (RR6). It has a pseudo-$R^2$ of 69%. For context, readers can find a similar cross-classified model using RR6 as an effect size in S1 Table.

The three presented models agree that when the frame is held at the "individual" level, mail mode and mixed-mode without face-to-face in the mix have significantly lower NBI than single-mode face-to-face. According to Model II, when the frame is held at the "individual" level, every mode is better in terms of nonresponse bias than pure face-to-face. When one holds the frame at the "nonindividual" level, the only mode bearing significantly different results than face-to-face is the mail mode; that mode used with a nonindividual frame is connected to a massive and significant increase in the NBI (This is the strongest effect observed in this study.) This problematic combination of mail and address frame in the survey design was most used in France (16 out of 28 cases). One should bear in mind that there are far fewer cases of mail and mixed-mode (no face-to-face) surveys connected to the household/address frame than to the frame of individuals: 28 to 73 for mail and 3 to 49 for mixed-mode. The ESS has significantly lower NBI than the ISSP and also more experience possibly gathered by the institution results in slightly lower bias. Model III, estimating partial effects of the mode, bears very similar results. Additionally, in this model, a higher RR is connected to lower bias.

## Discussion and conclusion

This study primarily aims to assess the relationship between survey mode and nonresponse bias. The choice of mode is one of the most crucial decisions when designing survey research. Whereas the existing opportunities usually impose the type of frame used and the sampling design, the mode choice is the compromise between methodological stance and possible budget. The most widespread belief is that the more expensive the mode, the better; and for cross-cut surveys, the face-to-face mode is usually the most expensive (It may not be true for poorly designed mixed-mode surveys.) While having the highest RRs from the analyzed modes, face-to-face does not have the lowest bias. The models presented here suggest that at least mail mode and mixed-mode surveys (without face-to-face) resulted in significantly smaller nonresponse bias than single mode face-to-face surveys when the frame of individuals was used. For mail surveys, this result was probably obtained—according to estimates present in the literature—with a much lower cost than face-to-face surveys. For mixed-mode surveys (without face-to-face), if they were designed properly, this remarkable result was also achieved with a lower cost than face-to-face surveys. These forms of mixed-mode surveys using different combinations of mail, web, and CATI were applied mainly in Nordic countries, such as Denmark (in 15 ISSP waves), Norway (10), Finland (8), and Iceland (5). Models II and III found that self-administered surveys, solo or mixed with face-to-face, also resulted in significantly lower NBI when the frame of individuals was used. In a scenario when sampling design was based on other types of frames, only mail mode was connected to higher NBI than face-to-face. It all suggests that the switch from face-to-face to other modes should be considered a great opportunity to raise research quality, even considering possible risks, mainly some kinds of mode effects [119].

All these results might mean that face-to-face surveys may not fully deserve the reverence they are held in based mostly upon the high RRs achieved, especially if we consider that this

**Table 2. Cross-classified meta-regression models estimating the relationship between survey characteristics and NBI.**

| | Model I | | | | | | | | Model II | | | | | | | | Model III | | | | | | | |
|---|---|---|---|---|---|---|---|---|---|---|---|---|---|---|---|---|---|---|---|---|---|---|---|---|
| | coef | se | tval | df | pval | ci.lb | ci.ub | | coef | se | tval | df | pval | ci.lb | ci.ub | | coef | se | tval | df | pval | ci.lb | ci.ub | |
| Intercept | 0.034 | 0.003 | 10.843 | 24 | <.001 | 0.028 | 0.041 | *** | 0.051 | 0.005 | 11.131 | 22 | <.001 | 0.041 | 0.060 | *** | 0.066 | 0.007 | 9.302 | 21 | <.001 | 0.051 | 0.081 | *** |
| Mixed-mode (with f2f) | -0.005 | 0.005 | -0.944 | 783 | 0.346 | -0.014 | 0.005 | | -0.010 | 0.005 | -1.996 | 781 | 0.046 | -0.021 | 0.000 | * | -0.010 | 0.005 | -2.013 | 780 | 0.044 | -0.021 | 0.000 | * |
| Mail mode | -0.011 | 0.004 | -2.475 | 783 | 0.014 | -0.020 | -0.002 | * | -0.019 | 0.005 | -3.938 | 781 | <.001 | -0.028 | -0.009 | *** | -0.019 | 0.005 | -4.086 | 780 | <.001 | -0.029 | -0.010 | *** |
| Mixed-mode (no f2f) | -0.012 | 0.005 | -2.525 | 783 | 0.012 | -0.021 | -0.003 | * | -0.017 | 0.005 | -3.607 | 781 | <.001 | -0.027 | -0.008 | *** | -0.020 | 0.005 | -4.006 | 780 | <.001 | -0.029 | -0.010 | *** |
| Nonindividual frame | 0.005 | 0.003 | 1.426 | 783 | 0.154 | -0.002 | 0.012 | | 0.001 | 0.003 | 0.367 | 781 | 0.714 | -0.005 | 0.008 | | 0.002 | 0.003 | 0.502 | 780 | 0.616 | -0.005 | 0.008 | |
| Mixed-mode (with f2f)*Nonindividual frame | 0.005 | 0.007 | 0.763 | 783 | 0.446 | -0.009 | 0.019 | | 0.008 | 0.007 | 1.147 | 781 | 0.252 | -0.006 | 0.022 | | 0.009 | 0.007 | 1.243 | 780 | 0.214 | -0.005 | 0.023 | |
| Mail mode*Nonindividual frame | 0.067 | 0.007 | 9.601 | 783 | <.001 | 0.053 | 0.081 | *** | 0.070 | 0.007 | 10.053 | 781 | <.001 | 0.056 | 0.083 | *** | 0.065 | 0.007 | 9.196 | 780 | <.001 | 0.051 | 0.079 | *** |
| Mixed-mode (no f2f)*Nonindividual frame | 0.011 | 0.012 | 0.866 | 783 | 0.387 | -0.013 | 0.035 | | 0.009 | 0.012 | 0.744 | 781 | 0.457 | -0.015 | 0.033 | | 0.009 | 0.012 | 0.760 | 780 | 0.447 | -0.014 | 0.033 | |
| ESS | | | | | | | | | -0.020 | 0.004 | -5.310 | 22 | <.001 | -0.028 | -0.012 | *** | -0.020 | 0.004 | -5.377 | 21 | <.001 | -0.027 | -0.012 | *** |
| Repetitions | | | | | | | | | -0.001 | 0.000 | -3.328 | 781 | 0.001 | -0.001 | 0.000 | *** | -0.001 | 0.000 | -3.818 | 780 | <.001 | -0.001 | 0.000 | *** |
| Response rate (RR6) | | | | | | | | | | | | | | | | | -0.025 | 0.009 | -2.823 | 780 | 0.005 | -0.043 | -0.008 | ** |
| Pseudo-R² | 0.58 | | | | | | | | 0.67 | | | | | | | | 0.69 | | | | | | | |

\* p<0.05

\*\* p<0.01

\*\*\* p<0.001

mode is also a victim of possible measurement issues, due mainly to the interviewer effects [120]. This finding is particularly optimistic in the context of limitations imposed by the COVID-19 pandemic, beginning in 2020, which made doing face-to-face research extremely hard. While the health concerns are in most cases lower at the time of writing this paper, changes in people's attitudes may be more long-lasting.

On the other hand, it must be stated that, compared especially to mail and CATI modes, direct interviewer involvement enables the collection of a lot more paradata, which is highly important in survey methodology research. This is an undeniable loss produced by the mode switch. Another disadvantage of using non-face-to-face modes concerns the interaction between the mail mode and the nonindividual frame (household, address, or area probability sampling). It resulted in a significant rise of NBI, probably because the within-household selection of the final respondent needed to be done by household members. They may not have understood the written instructions or might not have found these instructions that important when no interviewer was present to guide and motivate them to do it properly. This problematic situation was most common in the ISSP in France (16 out of 28 cases in the database). Lobbying by the research community to establish individual frames in countries where it is not present is highly advisable. Another factor resulting in lower bias was the (proxy of) institutional experience; doing more surveys in one country and one program resulted in lower NBI, which means that people do learn and improve their work over time. In addition, the ESS, according to the univocal assessment of the survey methodologist, being more methodologically "diligent" than the ISSP, had smaller NBI values.

This study's main limitation is that the survey modes in the analyzed programs were not assigned randomly to country and wave. Therefore, there is a possibility that behind all the models, there is one important hidden variable: the survey climate of the country [121 p815–17]. It was at least partially accounted for by using random intercepts for every country, but this solution is only viable if the survey climate is constant over time. This statement is probably false at least in some countries. The best way of overcoming this issue is through experimental research, yet it is very costly and impossible to conduct on a scale such as was used in this study. Of 54 meta-analyses in survey methodology listed by Čehovin, Bošnjak, and Lozar Manfreda, 28, more than half, used observational data [98]. Therefore, this limitation is real but very common in the field. The method of alleviating this limitation at least a little, used here, was the inclusion of ESS data. While ESS surveys used the single face-to-face mode only, they provided the possibility of comparing the results of surveys using this mode with ISSP surveys using another mode, but done in the same country and the same period. Unfortunately, it could only be done for European countries, thereby excluding examples of continuous use of the mail mode in New Zealand (14), Australia (13), and Canada (9). Another limitation comes from the impossibility, in the context of the ISSP data, of using the previously described "strict method" of NBI calculation—that is, controlling for the sex of the respondent's partner. A previous study suggests that this approach does not carry much risk [25]. The last limitation worth mentioning here is the fact that this paper covers modes and approaches used mostly in large comparative survey programs, which are relatively well funded, based on international collaboration, and usually academically and/or methodologically focused. For researchers dealing with smaller surveys, usually local, or especially commercially focused, and who are not used to pursuing full statistical representativeness, the data presented here may be, unfortunately, of small importance; the author is fully aware of this limitation.

The deluge of survey data gathered over recent years makes it nearly impossible to utilize their full analytical potential. In many cases funded from public sources, the surveying process should be used with care, and sparingly. Moreover, the inflation of surveys is clearly

detrimental to the surveying climate. Therefore, the method of meta-analysis in survey methodology research, while currently gaining in popularity, deserves to be utilized much more.

## Supporting information

**S1 File. PRISMA checklist.** From: [74].
(DOCX)

**S1 Table. Two-level meta-regression model using NBI as an effect size and three-level meta-regression model using RR6 as an effect size.**
(CSV)

## Acknowledgments

I would like to thank Piotr Jabkowski and Kamila Szymkowiak for their help in this research.

## Author Contributions

**Conceptualization:** Adam Rybak.

**Data curation:** Adam Rybak.

**Formal analysis:** Adam Rybak.

**Funding acquisition:** Adam Rybak.

**Investigation:** Adam Rybak.

**Methodology:** Adam Rybak.

**Project administration:** Adam Rybak.

**Resources:** Adam Rybak.

**Visualization:** Adam Rybak.

**Writing – original draft:** Adam Rybak.

**Writing – review & editing:** Adam Rybak.

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
