## [Decision Letter · Decision Letter 0]

23 Jun 2022

PONE-D-22-10734Survey mode and nonresponse bias: a meta-analysis based on the data from ISSP 1996–2018 and ESS rounds 1 to 9PLOS ONE

Dear Dr. Rybak,

Thank you for submitting your manuscript to PLOS ONE. After careful consideration, we feel that it has merit but does not fully meet PLOS ONE’s publication criteria as it currently stands. Therefore, we invite you to submit a revised version of the manuscript that addresses the points raised during the review process.

We thank authors for their manuscript on a very important topic of biases. The current version of manuscript, however, have several issues requiring a major revision. First, a more theoretical foundation should be provided; second, more clarification on models and in interpretation is needed. We encourage authors to incorporate reviews' suggestions, corrections, and feedback.

We look forward to receiving your revised manuscript.

Kind regards,

Olga Scrivner, PhD

Academic Editor

PLOS ONE

Journal Requirements:

Reviewers' comments:

Reviewer's Responses to Questions

**Comments to the Author**

1. Is the manuscript technically sound, and do the data support the conclusions?

Reviewer #1: Yes

Reviewer #2: Partly

Reviewer #3: No

2. Has the statistical analysis been performed appropriately and rigorously? 

Reviewer #1: No

Reviewer #2: No

Reviewer #3: Yes

3. Have the authors made all data underlying the findings in their manuscript fully available?

Reviewer #1: Yes

Reviewer #2: Yes

Reviewer #3: Yes

4. Is the manuscript presented in an intelligible fashion and written in standard English?

Reviewer #1: Yes

Reviewer #2: Yes

Reviewer #3: Yes

5. Review Comments to the Author

Reviewer #1: This study should include the following discussion on nonresponse bias (Halbesleben & Whitman, 2013):

1. Comparison of the sample population

2. Follow-up analysis

3. Wave analysis

4. Passive/Active nonresponse analysis (focus groups)

5. Interest level analysis

6. Benchmarking

7. Replication

This study utilizes the following as suggested by Johnson and Wislar (2012):

1. Analysis using data in sampling frame

2. Comparisons with survey from other sources

3. Analysis using external data sources

4. Analysis of paradata

The inclusion of a technological survey using computer audio (non-video) should be explored.

Reviewer #2: The author analyses nonresponse bias approached by the “internal criteria of representativeness”.

The nonresponse bias indicator (NBI) is given by the difference between the surveys proportion of

females in two-person households from 0.5. The author analysis the effect of mode on NBI

controlling for the sampling frame, interaction of sampling frame with mode, year of fieldwork,

survey experience in a country, and an indicator for ESS surveys.

The analysis includes two-level random intercept models and eight different models for the metaanalysis. Here, the definition of the third level is varied, and the models include and exclude RR6. The

author finds significant mode effects.

The topic is highly relevant. But I have some concerns about the method that should be addressed in

a revised version of the manuscript.

1. As NBI, the share of women in two-person households is considered, which according to the

theory should be 50%. An expectation of 50% only makes sense if two-person households are

exclusively heterosexual couples. This is not plausible because there are many living

arrangements. The assumption that there are equal numbers of homosexual female and

male couples is made here in a side note, but is essential. This needs to be emphasized more.

It would also help to explain how the mode comparisons are affected if this assumption is

not met.

2. Page 5: you state that Kohler argues that the internal criterion of representativeness is not

affected by item nonresponse. This is surprising and I would like to see some explanation of

the rationale in the manuscript.

3. The article explicitly considers only nonresponse and excludes other sources of error.

However, measurement errors can be very relevant here and can never be excluded when it

comes to survey data. And these can be related to the mode and bias the analysis. For

example, Felderer, Kirchner, and Kreuter (2019) find measurement error even for sociodemographic characteristics. The surveys have all been different and questionnaire effects

cannot be ruled out either. This should be discussed more in the limitations.

4. I’m not sure about the expression of the standard error of the NBI. Intuitively I would think

that the SE is \\sqrt{prop_fem*/1-prop_fem/n}. Why is it not? Please add a reference or

derivation. I’m not sure where the SE is used in the manuscript anyway.

5. The ESS is only conducted face-to-face and only in Europe. To what extent does it help to add

the ESS data here? ESS, mode and country are highly correlated. Is it not possible that the

ESS variable picks up parts of the mode effect (directly and indirectly being correlated with

country? I would like to see the analysis without controlling for ESS.

6. To understand the data better, I would like to see the face-to-face columns split by ESS and

ISSP in table 1.

7. The values of NBI are somewhere between 0 and 0.5. How is this reflected in the model?

In my opinion, there are too many models presented. I would prefer presenting either the clustering

of countries by the gdp or region, not both.

I am not clear what the random intercept two-level model looks like. Is it in columns 4 and 8 in the

table? Why is it needed here? What is the added value of figures 2 and 3, which if I understand

correctly are based on the model? Why is this presented if the meta-analysis is superior? Please

either elaborate the added value or delete parts of the analysis.

The meta-models are surprisingly robust when RR6 is added. But this is itself significant. How can this

be even though the variables in the model explain RR6?

Knowing whether gender is correctly represented in the group of two-person households is certainly

helpful. But what does it teach us about the survey as a whole? It may be that two-person

households are completely misrepresented or that other variables are affected by nonresponse bias.

This should be discussed more in the limitations.

Minor

To argue with changes due to the covid pandemic in the abstract is misleading. The data were

collected before the pandemic. Changes in fieldwork by Covid 19 prove the importance of the study,

but should rather be part of the introduction than the abstract.

Page 2, line 28: implementation of the survey

Page 4, line 67: the abbreviation NB needs to be introduced

Page 13. Line 273: I find this confusing. Please clarify when to use fixed and random effect models

and add a reference

ESS: on page 4 you state that it is fully face-to-fact while later you claim that it is

Reviewer #3: In his work, the author analyzes the effect of survey characteristics, especially the survey mode, on nonresponse bias. For this purpose, different waves of the ISSP and the ESS are analyzed and quantified to an overall estimate of the mode effect. The study concludes that mail and mixed-mode surveys are superior to interviewer-administered face-to-face data collection in regards to a reduction of nonresponse bias.

This paper is well written and deals with a highly relevant research question (possible effects of the data collection mode on nonresponse bias). Although the article is well written, there are some serious problems in regards to the research design and the theoretical foundation of the statistical models:

First, the research design of the present study is not suitable for ruling out alternative explanations of the variation of the dependent variable. This is because, the meta-analysis presented here summarizes non-experimental data sets. The original intention of “meta-analysis” is to increase the sample size and statistical efficiency by compiling and quantifying a series of (experimental) studies using very similar research designs and measures of key variables. However, in this study there is confounding of survey study (ESS versus ISSP; European Countries versus countries from all continents) and survey mode (face-to-face versus mixed mode designs). Therefore, the causal effect of the data collection mode on nonresponse bias cannot be identified because alternative hypotheses cannot be ruled out with the current research design.

Second, main theoretical terms and concepts are not consistently and correctly used. Especially, “survey representativeness” or “sample quality” is equated to “nonresponse bias” (see introduction or discussion part of the paper). Undoubtedly, nonresponse bias threatens the representativeness of a survey. However, “representativeness” is a multidimensional concept that is also influenced by other systematic error sources, such as coverage error or sampling bias. Therefore, the terms should not be used synonymously. Furthermore, the measure of nonresponse bias used in this study is very limited and focuses only on one variable (see page 10). However, nonresponse bias is strongly topic or variable specific. Besides the gender ratio, nonresponse bias of other variables of interest should have been explored and limitations of the indicator used in this work should have been discussed (such as possible reporting error that could bias the indicator used).

Third, I think that the paper needs a stronger theoretical foundation instead of giving a descriptive report of empirical results. Most covariates in the regression models are lacking a clear theoretical foundation (see Table 2). I cannot see a clear theoretical motivation for the choice of the moderator variables that are included in the statistical models. Are these variables confounders, mediators or collider variables? In the later case, it would not be advisable to control these variables. In an observational study aiming at estimation of the causal effect of a treatment variable (survey mode) on an outcome (nonresponse bias), the underlying causal model should be made explicit in order to decide which variables should be controlled for and which not. For example, how is the “survey frame” or the “year of the survey” related to the treatment variable and how is it supposed to affect nonresponse bias? I recommend to elaborate the theoretical part and derive clear hypotheses how the variables in the regression model affect the “survey mode – nonresponse bias” link.

6. PLOS authors have the option to publish the peer review history of their article (what does this mean?). If published, this will include your full peer review and any attached files.

Reviewer #1: No

Reviewer #2: No

Reviewer #3: No

---

## [Author Response · Author response to Decision Letter 0]

5 Nov 2022

Responses to Reviewers of the manuscript titled "Survey mode and nonresponse bias: a meta-analysis based on the data from ISSP 1996–2018 and ESS rounds 1 to 9"

To all Reviewers:

While considering the statistical suggestions I received in the revisions, I found some errors in standard error values in the database. I have corrected them and repeated every statistical procedure. It resulted in slight changes in the models but has not substantially changed the results.

I used your suggestion also to change the model specification and their number – currently, only three are presented in the body of manuscript, and two are in supplementary materials.

I used the opportunity of creating a new, corrected database and R script to upload them to the external repository instead of supplementary materials.

Response to Reviewer 1:

1. Follow-up analysis, passive and active nonresponse analysis and interest-level analysis can be effectively done only by the institution that has conducted or coordinated the survey because access to non-anonymized data about sample members is needed. The inclusion of such procedures usually is not reported in the survey methods reports.

2. Wave analysis is not applicable to the analysis of a large body of surveys from comparative programs with fieldwork organization hugely differing between countries. The fieldwork period is not identical even between ESS countries in each wave, which creates a problem with creating a standardized metric, and it is even more problematic with comparing early and late respondents between different survey modes.

3. Full replication of the whole survey of the country's population is usually not considered a viable solution to nonresponse bias assessment, because such endeavor’s financial cost is hard to justify.

4. Benchmarking is not applicable to "substantive" variables in analyses similar to those presented in this manuscript because topics covered in questionnaires differ between waves and programs. Therefore, multiple sources of differing quality and credibility are required as a benchmark.

Benchmarking based on demographic data comparison between one survey, and another depends on the decision, which of the two should be treated as a golden standard. Therefore, it has a very limited usage.

5. Comparison between the demographic composition of the net sample and official statistics is mentioned in the manuscript in the "nonresponse bias" section. Its main issue is that the quality of the official statistics may differ between countries as much as survey quality.

I added the reference (Halbesleben & Whitman, 2013) to the fragment describing the "external criteria of representativeness" – a comparison of demographic composition between respondents and the country's official statistics.

I added the following fragment at the end of the nonresponse bias section: "There are some additional possibilities for assessing the quality of the survey in terms of the nonresponse bias. One could use the gross sample information if it consists of individuals with known characteristics, but this is possible only in the case of individual samples (Johnson and Wislar, 2012). Additionally, nonrespondents could be targeted by very short follow-up questionnaires, as in ESS, but such follow-up may also be vulnerable to nonresponse. A similar vulnerability is shared by method comparing early and late respondents or individuals who needed the conversion procedures with the rest of the respondents (Halbesleben & Whitman, 2013). Such assessment methods are only possible for the secondary data analysis, when required information are easily accessible to external researchers, which is rarely the case."

Response to Reviewer 2:

1. I added these passages to address the issue of the inability to control for the same-sex couples: "For ESS rounds 1-8 and European Quality of Life Survey rounds 1-4, where controlling for same-sex couples is possible, the correlation between indicators using strict (with control) and lenient (as in this paper) approaches to asses NB was 0.96 and 0.93 respectively (Jabkowski, Cichocki, and Kołczyńska 2021). The inability to use a strict approach is an important limitation of this study."

2. I made the mistake of simply repeating Kohler's statement about criterion not being affected by item nonresponse, which obviously applied only to the surveys done with the participation of an interviewer. I added following sentence to address this: "(unfortunately only for face-to-face surveys, where gender is in most cases assessed by the interviewer)".

3. Focusing only on the nonresponse bias in my paper is a deliberate decision. I currently do not possess any data that make doing measurement error assessment at this scale possible. I have the NBI indicator based on the external UN demographic database in another dataset, but I find using it extremely problematic because of huge differences in official statistics quality between world regions, which is stressed in my manuscript.

Felderer, Kirchner, and Kreuter (2019) do not find measurement error for basic demographic characteristics, as used in my paper, only for those more prone to social desirability error, such as income or employment status. They write: "Not surprisingly, there is virtually no measurement error bias for gender in either mode (…) As expected, there is virtually no relative measurement error bias in mean age for either mode (…) there is no evidence for significant measurement error in any of the age categories, with relative measurement error biases being very close to zero". Additionally, they analyze data from only one country and with a linkage to administrative data for every respondent, which give them validation possibilities out of my reach.

As for the questionnaire (or topic) effects, in current models, they are at least partially accounted for by the random intercepts for waves, which shares the same (translated) questionnaires. Attitudes toward topics addressed in the questionnaires may differ between participating countries, but this is impossible to control on such a scale. Additionally, my paper is clearly focused on nonresponse bias. Total Survey Error paradigm made it possible to estimate different errors separately, and I am trying to do so for nonresponse.

4. I made the mistake of placing an explanation for the standard error calculation in a different part of the manuscript. To correct it, I repeated it just after the equation: "The standard error of this effect size is calculated based on the "true" gender proportion of 0.5". That is why prop_fem*(1-prop_fem)=0.5*(1-0.5)=0.5*0.5=0.25, as Kohler does it in the original paper. The inclusion of the equation is important, because it is clearly not obvious.

5. Inclusion of the "survey program" variable is important, because ESS and ISSP differ in much more areas than the used mode. I added references to papers describing these differences. I attach models without ESS data below to address the possibility of me choosing observations deliberately (I hope the output structure is not affected by the submission system). Code for models based on this subset of data is included in the R script for this paper, for readers information.

The model with "repetitions" variable included, ISSP only:

## Mixed-Effects Model (k = 572; tau^2 estimator: REML)

## 

## logLik deviance AIC BIC AICc 

## 1143.7383 -2287.4766 -2267.4766 -2224.1438 -2267.0780 

## 

## tau^2 (estimated amount of residual heterogeneity): 0.0002 (SE = 0.0001)

## tau (square root of estimated tau^2 value): 0.0126

## I^2 (residual heterogeneity / unaccounted variability): 17.40%

## H^2 (unaccounted variability / sampling variability): 1.21

## R^2 (amount of heterogeneity accounted for): 67.71%

## 

## Test for Residual Heterogeneity:

## QE(df = 563) = 656.0606, p-val = 0.0040

## 

## Test of Moderators (coefficients 2:9):

## F(df1 = 8, df2 = 563) = 26.9885, p-val < .0001

## 

## Model Results:

## 

## estimate se tval df pval ci.lb ci.ub 

## intrcpt 0.0437 0.0048 9.1516 563 <.0001 0.0343 0.0530 *** 

## modMixed-mode (with f2f) -0.0058 0.0054 -1.0858 563 0.2780 -0.0164 0.0047 

## modMail mode -0.0116 0.0049 -2.3640 563 0.0184 -0.0212 -0.0020 * 

## modMixed-mode (no f2f) -0.0127 0.0053 -2.3976 563 0.0168 -0.0230 -0.0023 * 

## fraNonindividual frame 0.0076 0.0044 1.7314 563 0.0839 -0.0010 0.0162 . 

## rep -0.0006 0.0002 -3.2584 563 0.0012 -0.0010 -0.0002 ** 

## modMixed-mode (with f2f):fraNonindividual frame -0.0016 0.0074 -0.2115 563 0.8326 -0.0161 0.0130 

## modMail mode:fraNonindividual frame 0.0663 0.0072 9.1598 563 <.0001 0.0521 0.0806 *** 

## modMixed-mode (no f2f):fraNonindividual frame 0.0082 0.0137 0.6022 563 0.5473 -0.0186 0.0351 

Model without "repetitions" variable included, ISSP only:

## Mixed-Effects Model (k = 572; tau^2 estimator: REML)

## 

## logLik deviance AIC BIC AICc 

## 1141.2511 -2282.5022 -2264.5022 -2225.4867 -2264.1773 

## 

## tau^2 (estimated amount of residual heterogeneity): 0.0002 (SE = 0.0001)

## tau (square root of estimated tau^2 value): 0.0131

## I^2 (residual heterogeneity / unaccounted variability): 18.54%

## H^2 (unaccounted variability / sampling variability): 1.23

## R^2 (amount of heterogeneity accounted for): 65.12%

## 

## Test for Residual Heterogeneity:

## QE(df = 564) = 668.0152, p-val = 0.0016

## 

## Test of Moderators (coefficients 2:8):

## F(df1 = 7, df2 = 564) = 28.7200, p-val < .0001

## 

## Model Results:

## 

## estimate se tval df pval ci.lb ci.ub 

## intrcpt 0.0343 0.0038 8.9528 564 <.0001 0.0268 0.0418 *** 

## modMixed-mode (with f2f) -0.0063 0.0054 -1.1592 564 0.2469 -0.0170 0.0044 

## modMail mode -0.0099 0.0049 -2.0258 564 0.0433 -0.0196 -0.0003 * 

## modMixed-mode (no f2f) -0.0122 0.0053 -2.2836 564 0.0228 -0.0226 -0.0017 * 

## fraNonindividual frame 0.0094 0.0044 2.1452 564 0.0324 0.0008 0.0180 * 

## modMixed-mode (with f2f):fraNonindividual frame -0.0030 0.0075 -0.4003 564 0.6891 -0.0176 0.0117 

## modMail mode:fraNonindividual frame 0.0642 0.0073 8.8114 564 <.0001 0.0499 0.0785 *** 

## modMixed-mode (no f2f):fraNonindividual frame 0.0089 0.0139 0.6409 564 0.5218 -0.0183 0.0361 

6. I added the ESS survey numbers separately in the table.

7. Values of NBI are between 0 to 0.29, and model coefficients correspond with these values. Because of your suggestions, I decided to redesign the models. Currently, there are only 3 cross-classified models, with surveys nested simultaneously in countries and waves, with three different sets of moderators (simple, extended, and with partial effects). Additionally, I included two models: a two-level meta-regression model using NBI as an effect size and a three-level meta-regression model using RR6 as an effect size in the supplementary materials, for readers that do not wish to use supplemented R script.

8. While for one or a couple of studies, there are sometimes possibilities of using much more sophisticated assessments of unit nonresponse bias, for a couple hundred of surveys internal criterion is the most useful. Apart of the many papers I cited in the manuscript, that use this measure, also (Koch et al. 2014) used it to assess ESS sample quality: "This internal criterion for sample quality has the advantage that the same fixed benchmark (50%) can be applied to all countries. Furthermore, problems as regards the comparability of measurement do not compromise the analyses. On the downside, we have to restrict the analyses to one survey estimate (gender) and to a subgroup of the sample (respondents from gender heterogeneous couples".

Additionally, I added following sentences for clarification: "Additionally, it must be stressed that, in reality, nonresponse bias is a property of a specific variable. It can have different sizes for different survey questions. Because of that, the measure used here must be treated only as a proxy."

9. While I treated the abstract as a place to advertise the usefulness of a paper for possible reader, and that made me include pandemic consequences in it, following your suggestion I changed the sentence to: " Together with other unpredictable events occurring in the world today, this poses a challenge: the necessity to accelerate a switch from face-to-face data collection to different modes, which were usually considered to result in lower response rates. "

10. As far as I could find, I have written that ESS uses PAPI and CAPI, not CATI.

Response to reviewer 3:

1. The Previous manuscript included two mentions of limitations coming from the observational character of the data. I included two more, to stress this fact adequately. 

A meta-analysis is a statistical analysis of a large collection of results from primary studies to integrate their findings (Glass 1976). While meta-analysis origins come from analyzing experimental data, it uses other types of data, even in the area of medicine and health studies, e.g. for disease frequencies estimation, or contraceptive prevalence estimation (Barendregt et al. 2013; Haakenstad et al. 2022; Kyu et al. 2022; Vos et al. 2020). In the survey methodology field, using observational data for meta-analysis is much more frequent. A paper analyzing the usage of this method in the field reports 28 manuscripts using meta-analysis on nonexperimental data from 54 included, more than a half (Cehovin, Bosnjak, and Lozar Manfreda 2018). 

2. I included models based on different subsets of data in the R script for this manuscript, for readers to check if the results hold. I attach models based on ISSP only, and European-only data below (I hope the output structure is not affected by the submission system). The direction of results is very similar to one based on the full set. I used many possible methods to make this analysis as robust as possible. I specify them in the completely rewritten methods and results sections.

Model based on ISSP data only:

## Mixed-Effects Model (k = 572; tau^2 estimator: REML)

## 

## logLik deviance AIC BIC AICc 

## 1141.2511 -2282.5022 -2264.5022 -2225.4867 -2264.1773 

## 

## tau^2 (estimated amount of residual heterogeneity): 0.0002 (SE = 0.0001)

## tau (square root of estimated tau^2 value): 0.0131

## I^2 (residual heterogeneity / unaccounted variability): 18.54%

## H^2 (unaccounted variability / sampling variability): 1.23

## R^2 (amount of heterogeneity accounted for): 65.12%

## 

## Test for Residual Heterogeneity:

## QE(df = 564) = 668.0152, p-val = 0.0016

## 

## Test of Moderators (coefficients 2:8):

## F(df1 = 7, df2 = 564) = 28.7200, p-val < .0001

## 

## Model Results:

## 

## estimate se tval df pval ci.lb ci.ub 

## intrcpt 0.0343 0.0038 8.9528 564 <.0001 0.0268 0.0418 *** 

## modMixed-mode (with f2f) -0.0063 0.0054 -1.1592 564 0.2469 -0.0170 0.0044 

## modMail mode -0.0099 0.0049 -2.0258 564 0.0433 -0.0196 -0.0003 * 

## modMixed-mode (no f2f) -0.0122 0.0053 -2.2836 564 0.0228 -0.0226 -0.0017 * 

## fraNonindividual frame 0.0094 0.0044 2.1452 564 0.0324 0.0008 0.0180 * 

## modMixed-mode (with f2f):fraNonindividual frame -0.0030 0.0075 -0.4003 564 0.6891 -0.0176 0.0117 

## modMail mode:fraNonindividual frame 0.0642 0.0073 8.8114 564 <.0001 0.0499 0.0785 *** 

## modMixed-mode (no f2f):fraNonindividual frame 0.0089 0.0139 0.6409 564 0.5218 -0.0183 0.0361 

Model based on European data only:

## Multivariate Meta-Analysis Model (k = 564; method: REML)

## 

## logLik Deviance AIC BIC AICc 

## 1243.4970 -2486.9939 -2460.9939 -2404.8708 -2460.3198 

## 

## Variance Components:

## 

## estim sqrt nlvls fixed factor 

## sigma^2.1 0.0000 0.0036 32 no wave 

## sigma^2.2 0.0000 0.0061 30 no cntry 

## sigma^2.3 0.0000 0.0016 564 no surv 

## 

## Test for Residual Heterogeneity:

## QE(df = 554) = 575.4607, p-val = 0.2557

## 

## Test of Moderators (coefficients 2:10):

## F(df1 = 9, df2 = 22) = 15.6551, p-val < .0001

## 

## Model Results:

## 

## estimate se tval df pval ci.lb ci.ub 

## intrcpt 0.0480 0.0050 9.6354 20 <.0001 0.0376 0.0584 *** 

## modMixed-mode (with f2f) -0.0080 0.0055 -1.4492 554 0.1479 -0.0189 0.0028 

## modMail mode -0.0180 0.0053 -3.4107 554 0.0007 -0.0284 -0.0077 *** 

## modMixed-mode (no f2f) -0.0160 0.0049 -3.2632 554 0.0012 -0.0256 -0.0064 ** 

## fraNonindividual frame 0.0004 0.0032 0.1313 554 0.8956 -0.0059 0.0068 

## progESS -0.0184 0.0042 -4.4053 22 0.0002 -0.0271 -0.0097 *** 

## rep -0.0008 0.0002 -3.3265 554 0.0009 -0.0012 -0.0003 *** 

## modMixed-mode (with f2f):fraNonindividual frame 0.0095 0.0078 1.2294 554 0.2194 -0.0057 0.0248 

## modMail mode:fraNonindividual frame 0.0690 0.0073 9.4339 554 <.0001 0.0546 0.0833 *** 

## modMixed-mode (no f2f):fraNonindividual frame 0.0104 0.0118 0.8755 554 0.3817 -0.0129 0.0336 

3. I used "sample quality" and "representativeness" to avoid repetitions. It is worth mentioning, that Kohler uses the term "internal criteria of representativeness" in his paper, and (Koch et al. 2014) gives the report, that uses Kohlers measure for ESS data the title: "Assessing ESS sample quality by using external and internal criteria". However, following your suggestion, I removed such terms from discussion section of manuscript, while leaving them in the introductory section, where they correspond to the claims of other authors.

4. Following your suggestion, I added the sentence: "Additionally, it must be stressed that, in reality, nonresponse bias is a property of a specific variable. It can have different sizes for different survey questions. Because of that, the measure used here must be treated only as a proxy." I summarize the conclusions of (Jabkowski et al. 2021), and give reference for this paper, which extensively analyzes the possibilities of measuring nonresponse bias for a large body of surveys conducted in different countries. Felderer, Kirchner, and Kreuter (2019), a paper suggested by another Reviewer, inquire into the subject of misreporting demographic data. For simple variables, not prone to social desirability bias (as used in the internal criterion) they state: "Not surprisingly, there is virtually no measurement error bias for gender in either mode (…) As expected, there is virtually no relative measurement error bias in mean age for either mode (…) there is no evidence for significant measurement error in any of the age categories, with relative measurement error biases being very close to zero".

5. I have rewritten the methods section to add more foundations to the choice of moderators and address their status in the possible causal inference model: "Choice of moderators for the models was based on both state-of-the-art and statistical evaluation. Survey mode and frame constitute probably the most basic set of survey characteristics. The subject of mode was extensively described in the introductory parts of this paper. As for the sampling frame, the frame of individuals is usually considered better in terms of representativeness than other types of frames, primarily because of the burden connected to the process of within-household-selection of the respondent at the last sampling stage (Gaziano 2005; Jabkowski 2018; Menold 2014). Therefore, I used the binary variable individual/nonindividual (household/address, area probability sampling). To not overcomplicate the model, I ignored the possible impact of the within-household selection method on NB (Jabkowski 2017). Exploratory subgroup analysis and previous knowledge about possible issues connected with the aforementioned respondent selection not being supervised by the interviewer (as in e.g. mail mode) led to the conclusion, that also the interaction between those two moderators may be an important factor in explaining differences in NBI values (Reader can conduct all analyses mentioned in the article, and some more, using the R script available at …). Therefore mode, frame, and interaction between them were selected for the simplest model specification. The analyzed dataset consists of studies from two survey programs, that differ significantly in terms of funding and organization, (Bréchon 2009; Smith and Fu 2016) hence the inclusion of binary variable ESS/ISSP in the extended model. Institutional experience can influence outcome rates of a survey, (Stoop et al. 2010:102–13) and thus at least the upper boundary of possible nonresponse bias. I chose to include a moderator, that for each observation simply counts how many surveys were done previously in this country and program. This study, as stated previously, is based on observational data – like many meta-analyses in survey methodology (Cehovin et al. 2018). If one, however, would like to apply the categories of variables usually included in the causal inference models, then this study is focused chiefly on the effect of survey mode on nonresponse bias, and every candidate for the meta-regression moderator listed above should be treated as a confounder. Sampling frame, institutional experience, and survey program can usually have an impact on the choice of mode(s) and separately on the nonresponse bias. As for the last candidate for a moderator – the response rate (RR6) – its inclusion can be used to estimate the partial effects of survey modes holding RR constant. In the causal inference model, it would be a mediator. RR6 was included only in the last of the models presented in the results."

6. While I am clearly not placing my reasoning in the causal inference paradigm, which would be extremely interesting, and following your suggestions, I am thinking about conducting different studies using some kind of matching technique, I can specify what would be the role of my moderators in my model. It is shortly mentioned in the modified parts of the manuscript cited above, but here I would like to elaborate further. The sampling frame chosen for the specific survey impacts both the choice of mode (e.g. some forms of area probability sampling are not possible to conduct without at least face-to-face screening, a necessity to conduct within-household selection will discourage use of modes, where there is no interviewer to at least supervise the selection, etc.) and the nonresponse (burden of within household selection resulting in more refusals, possibility of using fully personalized communication while having access to individual register, etc.). The survey program similarly impacts the choice of modes (as ESS does not give any choice) and the nonresponse (e.g. because of substantial funding differences). Finally, "repetitions" will also impact both the choice of modes (previous experience may lead to innovations) and the nonresponse (a more experienced team will implement the chosen survey design better, at least usually). I have excluded the year variable from the models, both because it lacked strong theoretical foundations, and because of statistical evaluation.

---

## [Decision Letter · Decision Letter 1]

20 Jan 2023

PONE-D-22-10734R1Survey mode and nonresponse bias: a meta-analysis based on the data from ISSP 1996–2018 and ESS rounds 1 to 9PLOS ONE

Dear Dr. Rybak,

Thank you for submitting your manuscript to PLOS ONE. After careful consideration, we feel that it has merit but does not fully meet PLOS ONE’s publication criteria as it currently stands. Therefore, we invite you to submit a revised version of the manuscript that addresses the points raised during the review process.

We look forward to receiving your revised manuscript.

Kind regards,

Olga Scrivner, PhD

Academic Editor

PLOS ONE

Additional Editor Comments:

The manuscript has been improved since its initial submission. However, there are some major shortcomings pointed out by the reviewers that the author is encouraged to address.

Reviewers' comments:

Reviewer's Responses to Questions

**Comments to the Author**

1. If the authors have adequately addressed your comments raised in a previous round of review and you feel that this manuscript is now acceptable for publication, you may indicate that here to bypass the “Comments to the Author” section, enter your conflict of interest statement in the “Confidential to Editor” section, and submit your "Accept" recommendation.

Reviewer #2: (No Response)

Reviewer #4: (No Response)

2. Is the manuscript technically sound, and do the data support the conclusions?

Reviewer #2: Yes

Reviewer #4: Yes

3. Has the statistical analysis been performed appropriately and rigorously? 

Reviewer #2: I Don't Know

Reviewer #4: I Don't Know

4. Have the authors made all data underlying the findings in their manuscript fully available?

Reviewer #2: Yes

Reviewer #4: Yes

5. Is the manuscript presented in an intelligible fashion and written in standard English?

Reviewer #2: Yes

Reviewer #4: No

6. Review Comments to the Author

Reviewer #2: The authors are overall responsive to my comments and suggestions. I still see some issues that need to be solved:

The standard error of the effect size does still not convince me. The effect size is given as the absolute value of the difference between the observed proportion and the true proportion of 0.5.

The equation in the paper is just the standard error of a proportion of 0.5. It does only depend on n but not on the observed proportion in the survey at all. How can that be the standard error of the effect size?

Page 17, line 356: must be NBI instead of NDI

What is shown in figure 3? As I understand, these are the RR6 for the different modes. How can you conclude from looking at figure 2 and figure 3 that there is a “connection between realization and bias”?

Table 2: only the results for model 1 are shown

Page 24, lines 508 and 509: You can not interpret the significant of a main effect if there are interaction terms including this effect in the model

Reviewer #4: Please see uploaded document. The main issue with this paper is to clean up the writing. I pointed out some ideas for how this can be addressed. Generally, fewer acronyms would be good. The transition between the literature review and the results section is the area that needs the most work. But a few good transition sentences should suffice and I don't need to see it again.

7. PLOS authors have the option to publish the peer review history of their article (what does this mean?). If published, this will include your full peer review and any attached files.

Reviewer #2: No

Reviewer #4: No

---

## [Author Response · Author response to Decision Letter 1]

24 Feb 2023

I would like to thank the reviewers very much for suggesting many important improvements, which I have tried to implement in my manuscript. Additionally, below I present my responses to some issues.

To Reviewer #2

1. I have changed the part of my paper where I describe the chosen effect size to:

“NBI, being in this context a single study effect size, was calculated as: 

ESi = |propfem - 0.5|

where propfem is the proportion of females in a subsample of couples living together in two-person households and n is the size of a subsample of couples living together in two-person households (Kohler 2007). As the proportion on which the ESi is based is only an estimate and the true proportion of females in the defined subpopulation is already known to equal 0.5, the within-study variance may be calculated as: 

Vi = 0.25/ni

Where ni is the subsample size of the single study (Kohler 2007; Jabkowski and Cichocki 2019; Eckman and Koch 2019).”

If I used the proportion in the survey subsample as a numerator in the variance equation, the calculation would assign very low variance to the subsamples with a very high or low proportion of females (sometimes even 1 or 0). This approach does not make sense because this “wrong” proportion is often caused by the tiny size of the subsample, resulting from some traditional societies not having many two-person households of partners. It would assign these studies much higher weight. If we use the known proportion of 0.5, these studies have a lower weight, as intended.

If we use the equation for the variance of the difference between proportions V = (p1 * (1 - p1) / n1) + (p2 * (1 - p2) / n2), n1 will be the size of the country’s population “living together in…,” estimated by multiplying the country’s population above a certain age for a given year and proportion in the survey subsample. This would put numbers like a couple of millions in the denominator, making the first part of the equation nearly irrelevant. The second part of the equation would be the same as I described in the previous paragraph, with the same problems. Using Vi = 0.25/n, like the cited authors, is the best option. 

2. Looking at the forest plots, I do not conclude that there is a connection between RR6 and bias. I have slightly rewritten this paragraph to make it even clearer: “As we can see in figure 2, simple random-effects meta-analysis in subgroups shows that NBI values for mixed-mode surveys with face-to-face (mostly face-to-face with additional self-administered questionnaires) and for the mail mode clearly overlap with the estimate for single-mode face-to-face surveys. At this level, it is not possible to tell whether values of bias estimate for other mixed-mode surveys are smaller than for face-to-face. For context, we can see in figure 3 that the values of RR6 for modes using face-to-face surveys in any configuration clearly outperform other ones, which is in accordance with common expectations concerning this subject. Based on the presented literature, it was expected that the connection between realization and bias would be weak at most.”

3. It is true that I have treated the interpretation of insignificant interactions too lightly. I have rewritten parts of the Results and Discussion section to make it clearer that in this configuration of moderators you can only compare one frame type at a time, and I added a comparison model with the reference level of frame reverted mentioned in the text, and in the script.

4. Table 2 appears in such way, because the manuscript is converted to a PDF format. I think I sticks to the formatting guidelines; if not, I will discuss this topic with the editor. For the reviewers, I previously attached table 2 as a separate file and it is still available as an attachment.

To Reviewer #4

1. I changed all the NBs back to “nonresponse bias.” Therefore, now the only acronym which is not conventional in the field (along with RR, ESS, ISSP) is NBI (nonresponse bias indicator), which I am really convinced should stay in this abbreviated form to avoid constant repetition of such a long term.

2. I must admit that the quick step from the summary of the state of the art to the research questions was my intention, connected to the intention of creating a certain feeling in the reader (that the findings are confusing, that there are a lot of gaps in the knowledge, and that therefore this study is needed). However, following your advice, I expanded these two sentences into a full paragraph to make the transition smoother.

3. While local or commercial survey research has already mostly switched to web or phone designs, and in this area differences between PC and mobile are far more important than face-to-face vs. mail, these issues still seem essential in the big comparative projects focused on statistical representativeness. Not even the COVID-19 pandemic has forced behemoths like ESS or Eurobarometer to fully drop face-to-face. The only big program that switched to CATI in full in response to the restrictions was LAPOP (the AmericasBarometer), but it uses sophisticated quotas on household level and is therefore hard to consider methodologically driven. I added these sentences to the limitations section: “The last limitation worth mentioning here is the fact that this paper covers modes and approaches used mostly in large comparative survey programs, which are relatively well funded, based on international collaboration, and usually academically and/or methodologically focused. For researchers dealing with smaller surveys, usually local, or especially commercially focused, and who are not used to pursuing full statistical representativeness, the data presented here may be, unfortunately, of small importance. The author is fully aware of this limitation.”

4. I state the following in the nonresponse bias section: “Additionally, it must be stressed that nonresponse bias is a property of a specific variable. It can have different sizes for different survey questions. For this reason, the measure used here must be treated only as a proxy.” This whole section is devoted to explaining how hard it is to assess the bias, especially if no auxiliary data are present. I base my confidence in choosing the Kohler’s approach (even without controlling for same-sex couples and using weights) on the analysis by Jabkowski, Cichocki, and Kołczyńska (2021, doi: 10.1093/jssam/smab027) that focused clearly on the selection of bias indicators for this type of data and concluded that this method is optimal. The method, as I explain in my paper, is based on the proportion of genders in a defined subsample of two-person households of couples. As for the impact of different motivations and incentives for interviewers on the respondent selection process, Eckman and Koch (2019, doi:10.1093/poq/nfz012) used the Kohler measure to assess the bias coming from that source. Therefore, it may be considered covered in my analysis, because there is no motivation for interviewers to keep a certain proportion of genders in the specific subsample.

5. As for the “causal inference paradigm” terms (suggested by the previous reviewer), I have arbitrarily chosen to focus on the mode as a variable of main interest. In that situation, frame, program, and “repetitions” work clearly as confounders. The sampling frame chosen for the specific survey impacts both the choice of mode (e.g., some forms of area probability sampling are not possible to conduct without at least face-to-face screening; the necessity of conducting within-household selection will discourage the use of modes in which there is no interviewer to at least supervise the selection, etc.) and the nonresponse (once more, the burden of within-household selection, the possibility of using personalized communication while sticking to an individual frame, etc.). The survey program similarly impacts the choice of modes (as ESS does not give any choice) and the nonresponse (because of huge funding differences). Finally, “repetitions” will also impact both the choice of modes (previous experience may lead to innovations) and the nonresponse (A more experienced team will implement the chosen survey design better, at least usually.) RR6 is a mediator, as it is partially determined by the survey characteristics, and it also have some effect on nonresponse bias.

6. My data suggests that there is a significant relation between RR and NBI. The higher response rate, the lower the bias. It is visible in the model with the inclusion of RR6 as one of the moderators, and if one used it as a sole moderator in that model, the relationship would be even stronger. I have consciously omitted elaboration on this subject in my paper, because a reviewer from Survey Research Methods (where I sent the earliest version of this paper, containing the mentioned model) advised me that the question of pure relation between RR and bias was interesting 10 years ago and that I should focus on the subject of lowering the bias. The data and scripts are available and fully commented on by me. As I now see this subject not as important, I would prefer to leave it in the script, for the very interested researchers.

7. PAPI/CAPI are “pen & paper” and “interviewer with a laptop” variations of face-to-face. It does not make for a huge difference (which is why I do not account for it in my models), but for some researchers it is important, like the PC/mobile difference in CSAQ (while others even see PC and mobile as separate modes).

8. I am currently ending a project that maps survey designs in big, comparative, noncommercial survey projects around the globe, and the only one that used a large portion of phone interviewing was LAPOP. There is generally a problem with running statistically driven surveys by phone, because of issues with the frame (or especially because of the lack of it).

9. I have expanded the paragraph defining what an effect size is. It is mainly targeted to reduce possible confusion that readers used to standard measures used in clinical research and similar may feel. Usage of NBI as an effect size in meta-analysis was performed in a couple of papers that I cite. The paragraph now reads as follows: “Most commonly, meta-analysis is used to create proper statistical synthesis of results taken from many articles or clinical trials. In that situation, some typically used dependent variables, called effect sizes or treatment effects (Borenstein et al. 2009, 17), are usually computed to either statistically standardize the different measures used by different authors or comply with standard settings of experimental studies in medicine. Therefore, readers probably encountered two types of effect sizes: ‘those based on differences of two means and those based on differences of two proportions’ (Hartung, Knapp, and Sinha 2008, 13). However, these examples do not exhaust the usefulness of meta-analysis. On the contrary, “effect size is simply the size of anything that may be of interest” (Cumming 2012, 34), whether it is measured in exactly the same way for every study or recalculated to make it so. For example, in epidemiological studies, the true population mean, or proportion, may be of interest (Barendregt et al. 2013). Card proposed that the working definition of effect size could be “an index of the direction and magnitude of association between two variables” (Card 2012, 87), a definition that would include the measure used in this paper, if interpreted sensu largo. 

10. As for the paradata, it is important in the studies on the European Social Survey methodology on one side, and on the other, it is an important part of the responsive/adaptive/tailored survey design that has been implemented with some success, at least in the Netherlands.

11. I also implemented many minor corrections suggested by the reviewer.

---

## [Editor Report · Decision Letter 2]

2 Mar 2023

Survey mode and nonresponse bias: a meta-analysis based on the data from the International Social Survey Programme waves 1996–2018 and the European Social Survey rounds 1 to 9

PONE-D-22-10734R2

Dear Dr. Rybak,

We’re pleased to inform you that your manuscript has been judged scientifically suitable for publication and will be formally accepted for publication once it meets all outstanding technical requirements.

Kind regards,

Olga Scrivner, PhD

Academic Editor

PLOS ONE

Additional Editor Comments (optional): Thank you for addressing comments and updating your manuscript.

---

## [Editor Report · Acceptance letter]

6 Mar 2023

PONE-D-22-10734R2 

Survey mode and nonresponse bias: a meta-analysis based on the data from the International Social Survey Programme waves 1996–2018 and the European Social Survey rounds 1 to 9 

Dear Mr. Rybak:

I'm pleased to inform you that your manuscript has been deemed suitable for publication in PLOS ONE. Congratulations! Your manuscript is now with our production department. 

Kind regards, 

on behalf of

Dr. Olga Scrivner 

Academic Editor

PLOS ONE